# Transient circular dichroism and exciton spin dynamics in all-inorganic halide perovskites

Weijie Zhao [1,3], Rui Su[1,3], Yuqing Huang[1], Jinqi Wu[1], Chee Fai Fong [1], Jiangang Feng[1] & Qihua Xiong [1,2✉]

All-inorganic metal halides perovskites ($CsPbX_3$, X = Br or Cl) show strong excitonic and spin-orbital coupling effects, underpinning spin-selective excitonic transitions and therefore exhibiting great promise for spintronics and quantum-optics applications. Here we report spin-dependent optical nonlinearities in $CsPbX_3$ single crystals by using ultrafast pump-probe spectroscopy. Many-body interactions between spin-polarized excitons act like a pseudo-magnetic field and thus lift the degeneracy of spin states resulting in a photoinduced circular dichroism. Such spontaneous spin splitting between "spin-up" and "spin-down" excitons can be several tens of milli-electron volts under intense excitations. The exciton spin relaxation time is ~20 picoseconds at very low pump fluence, the longest reported in the metal halides perovskites family at room temperature. The dominant spin-flip mechanism is attributed to the electron-hole exchange interactions. Our results provide essential understandings towards realizing practical spintronics applications of perovskite semiconductors.

[1] Division of Physics and Applied Physics, School of Physical and Mathematical Sciences, Nanyang Technological University, Singapore 637371, Singapore. [2] State Key Laboratory of Low-Dimensional Quantum Physics and Department of Physics, Tsinghua University, Beijing 100084, China. [3] These authors contributed equally: Weijie Zhao, Rui Su. ✉email: qihua@ntu.edu.sg

Metal halide perovskites (MHPs) with general formula of ABX$_3$ (where A is an organic or inorganic cation, B = Pb, and X = Cl, Br, or I) are semiconducting materials with exceptional optical and electrical properties[1–11]. In the past few years, tremendous progress has been made for developing high-performance optoelectronic and photonic devices based on MHPs, such as solar cells[3], light-emitting diodes[8], and lasers[2,5,12,13], demonstrating great promise towards practical applications. Meanwhile, MHPs exhibit distinct spin-dependent physical properties arising from the strong spin-orbital coupling in the heavy Pb atoms[14,15]. The conduction band minimum (CBM) is mainly composed of the Pb 6$p$ orbitals, while the valence band maximum (VBM) consists of a hybridization of Pb 6$s$ ad Br 4$p$ orbitals with an overall $s$ symmetry[14,15]. The CBM and VBM both exhibit a double degeneracy with electrons and holes characterized by angular momentum $j_{e(h),z} = \pm 1/2$ (Fig. 1a)[15–17]. Consequently spin-dependent optical selection rules allow the efficient injection of spin-polarized carriers or excitons (Coulomb correlated electron–hole pairs with angular momentum $J_z = j_{e,z} + j_{h,z} = \pm 1$) by using circularly polarized lights[14,15,18–20]. Therefore, it is projected that the controlled manipulation of spin orientations of carriers or excitons in MHPs with external stimuli, like the magnetic and optical field, shall be exploited for future spintronic applications.

Recent experiments have revealed long carrier or exciton spin coherence at liquid-helium temperature[19,21], exotic magnetic field-induced spin-mixing effects[17], the Rashba effect[14–16,22] and long-range spin-funneling[23] in perovskite materials or their nanostructures, which have triggered extensive research interest on perovskite spintronics. However, the experimental observation and fundamental understanding of the spin-contrasting optical phenomena in MHPs are still at its nascent stage, especially, for all-inorganic MHPs (CsPbX$_3$). In contrast to the hybrid organic–inorganic perovskites, CsPbX$_3$ show distinct properties, such as excellent environmental stability and robust excitonic effects[7,12], and would be ideal for spin-optoelectronic and quantum-optic applications. For instance, Bose–Einstein condensation and lasing of exciton polaritons in the CsPbX$_3$ microcavities have been demonstrated at room temperature recently[7,12,24,25]. Introducing spin functionalities in such system will facilitate the development of spin-based quantum logic devices[26–30].

Here, we report the intriguing spin-contrasting phenomena of excitons in CsPbX$_3$ single crystals by using time- and polarization-resolved pump–probe spectroscopy. The CsPbBr$_3$, CsPbCl$_3$, and CsPbBr$_x$Cl$_{3-x}$ single-crystalline films are epitaxially grown on mica substrates by using the chemical vapor transport method ("Methods")[5,13,31]. The single-crystalline nature of CsPbX$_3$ thin films can be confirmed through detailed sample characterizations by using optical spectroscopy, scanning electron microscopy (SEM), high-resolution X-ray diffraction (XRD), and atomic force microscopy (AFM) shown in Supplementary Fig. 1. We systematically investigate nonlinear optical responses of excitons in CsPbX$_3$ as a function of excitation wavelength, pump fluences, and lattice temperatures. The photoinduced circular dichroism (CD) is observed in the form of a significant blueshift or a minor redshift of exciton absorption peak with co- or cross-circular pump–probe configurations, respectively. The underlying mechanism is attributed to the many-body interactions of spin-polarized excitons which give rise to a strong repulsive interaction among excitons with parallel spin and a much weaker attractive interaction among excitons with anti-parallel spin[32–35]. The photoinduced spin splitting is ~12 meV at a moderate excitation density (~9.4 μJ/cm$^2$). The magnitude of this spin splitting is almost 10 times larger than that reported by applying magnetic field (~1.3 meV at 10 teslas[21]). We further extract the exciton spin relaxation time to be up to ~20 ps under low excitation densities at room temperature. The mechanism that governs exciton spin relaxations in CsPbX$_3$ is the electron-hole exchange interaction, namely the Bir–Aronov–Pikus (BAP) mechanism[36].

## Results

**Spin-dependent optical nonlinearities.** The transient absorption (TA) spectra of CsPbBr$_3$ are recorded with differential transmissions, defined as $\Delta T/T = (T_{pump-on} - T_{pump-off})/T_{pump-off}$, at controlled delay times. We use a right (σ+) circularly polarized pump pulse resonant with the exciton peak (~2.4 eV) to selectively generate +1 excitons in the system. The consequent spin-dependent optical nonlinearities are probed by using broadband right or left (σ+ or σ−) circularly polarized light as demonstrated by the schematics in Fig. 1a. The time evolution of TA spectra with σ+ σ+ and σ+ σ− pump–probe configurations are shown in Fig. 1b, c, respectively, with a pump fluence of ~0.9 μJ/cm$^2$ at room temperature. In an exemplary spectrum at delay time of 0.3 ps in Fig. 1b (black curve), a significant pump-induced bleaching feature (PIB, as $\Delta T/T > 0$) dominates, while a weak pump-induced absorption feature (PIA, as $\Delta T/T < 0$) appears at energies

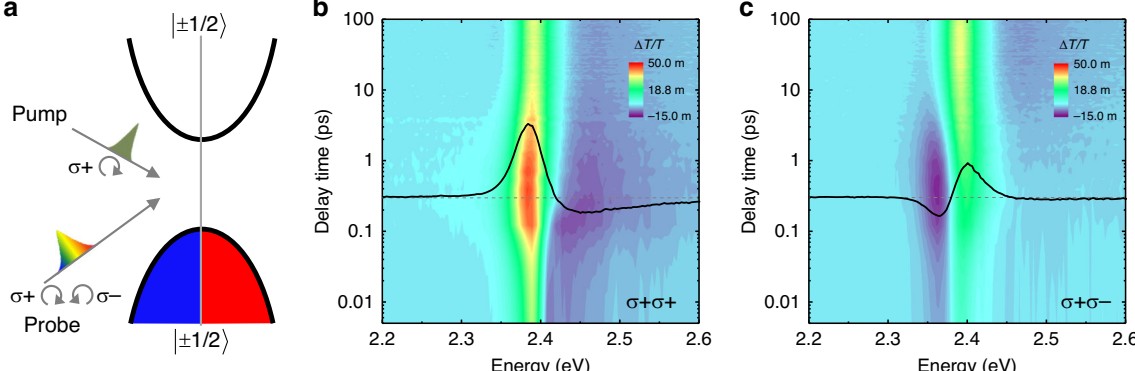

**Fig. 1 Spin-dependent optical nonlinearities in CsPbBr$_3$ at room temperature. a** Schematics of the resonant pumping and broadband-probing transient absorption (TA) spectroscopy and the electronic band structure of CsPbBr$_3$. The doubly degenerate spin states at the conduction band minimum or valence band maximum have angular momentum ±1/2. The laser beam with right (σ+) or left (σ−) circular polarization can generate exciton states with total angular momentum +1 or −1. **b, c** Time evolution of TA spectra of CsPbBr$_3$ obtained with co-circularly (σ+ σ+) and cross-circularly (σ+ σ−) polarized pump and probe pulses, respectively, at room temperature. The black curves in **b** and **c** show the TA spectra at a delay time of 0.3 ps, and the associated horizontal gray dashed lines are baselines with $\Delta T/T = 0$. The photon energy of pump beam is ~2.4 eV, which is in resonant with the exciton states. The pump fluence is ~0.9 μJ/cm$^2$.

higher than the exciton resonance. In contrast, the σ+ σ− TA spectrum at 0.3 ps in Fig. 1c show a reversed profile, in which the PIB/PIA feature appear at the high/low energy side with similar magnitudes, respectively. The difference between σ + σ+ and σ+ σ− TA spectra builds up quickly and reaches its maximum at ~0.3 ps after photoexcitation, and then decreases as a function of delay time. The TA spectra with σ− σ− and σ− σ+ pump–probe configurations were also measured as control experiments and show the same spin-dependent optical phenomena (see Supplementary Fig. 2). Identical σ+ σ+ and σ+ σ− TA spectra are observed at delay times longer than ~50 ps and coincide with those obtained by using linearly polarized pump and probe pulses (see Supplementary Fig. 3). The substrate effect from Mica are found to be minimal by checking $CsPbBr_3$ samples transferred on other substrates like sapphire wafers (see Supplementary Fig. 4). Such spin-dependent optical nonlinearities are observed for the first time in bulk all-inorganic MHPs. In the following, we first discuss the physical origins of the spin-contrasting TA spectra and then focus on the exciton spin relaxation mechanism.

**The photoinduced CD.** We extract the absorption spectra at various delay times shown in Fig. 2a and more results are referred to Supplementary Fig. 5. In Fig. 2a, the spectrum at $\Delta t = -2.0$ ps represents the linear absorption of the sample without pumping, which is compared with the nonlinear absorption spectra at 0.3 ps after photoexcitation. The exciton resonance shows a significant blueshift and reduction of oscillator strength in the σ+ σ+ spectrum. Following the incident σ+ pump pulse, coherent optical polarization will be initialized and subsequently transfers into an incoherent exciton population[37,38]. The polarization-population conversion process can be limited within ~0.2 ps in MHPs revealed by using coherent four-wave mixing spectroscopy[39]. Therefore, the reduction of exciton oscillator strength at 0.3 ps is due to the phase-space filling effect[40], while the blueshift is attributed to the inter-exciton repulsive interaction owing to the Pauli exclusion principle acting on the electronic constituents in the excitons[34,35].

On the contrary, a minor redshift of the absorption edge is observed in the σ+ σ− spectrum as indicated by the gray arrow in Fig. 2a. We can thus ascertain that the exciton peak, although not well-separated from the continuum band at room temperature, shows a redshift since other possible contributions (like the broadening effect) are very minor as to be discussed later. The attractive interaction between the +1 and −1 excitons created by the σ+ pump and σ− probe beams respectively, accounts for this

exciton redshift[32–34]. Moreover, we have noticed that the −1 exciton resonance loses little oscillator strength, which means that a small fraction of the injected excitons switches their spin orientation from +1 to −1 during the polarization-population conversion process[38]. In other words, the exciton spin injection efficiency is slightly below 100% in the sample.

The striking difference between σ+ σ+ and σ+ σ− absorption spectra indicates a photoinduced CD in $CsPbBr_3$. A general physical picture is proposed by using the schematics shown in Fig. 2b. When the pump and probe pulses have the same circular polarization (σ+ σ+), a strong absorption bleaching and significant blueshift of +1 exciton states lead to a slightly dispersive TA spectrum, which explains the TA spectra in Fig. 1b qualitatively. For the oppositely polarized pump and probe (σ+ σ−), the −1 exciton states present a peak redshift and a very minor absorption bleaching, thus leading to a more dispersive TA spectrum (Fig. 1c). Moreover, −1 excitons as the minority in the system would experience strong scattering from the majority +1 excitons, and subsequently the absorption linewidth would be broadened[38,41]. However, as evidenced by the tiny PIA signal at the high-energy side in σ+ σ− TA spectra at earlier delay times (Fig. 1c and Supplementary Fig. 3), optical nonlinearities introduced by the broadening effect is about one order of magnitude weaker than those from the phase-space filling and peak redshift.

Since only exciton population can give rise to a net absorption bleaching under resonant excitation, we can estimate the circular polarization degree ($P$) of the excitons by using the integrated σ+ σ+ and σ+ σ− TA spectra[42], denoted as $I_{++}$ and $I_{+-}$, respectively. Therefore, $P = (I_{++} - I_{+-})/(I_{++} + I_{+-})$. The circular polarization degrees at 0.3 ps as a function of pump fluence are presented in Fig. 2c. The polarization degree ranges from ~80 to ~90% for all excitation densities used in the measurements, suggesting a high spin injection efficiency in $CsPbBr_3$. Meanwhile, the energy splitting ($\Delta E$) between the +1 and −1 exciton states at 0.3 ps are shown in Fig. 2c (right axis) and Supplementary Fig. 5. A roughly linear increase of $\Delta E$ is observed as a function of pump fluence. At a moderate pump fluence of ~9.4 μJ/cm², the $\Delta E$ is ~12 meV, which is more prominent compared with those reported in GaAs-based quantum wells[32,33,41,43]. In addition, it is noteworthy that other effects arising from a population of non-polarized excitons, such as the short-range Coulomb interaction[40], band-filling (i.e., the Burstein–Moss effect)[44] and dielectric screening[43,45], may also introduce notable optical nonlinearities. Nevertheless, the contribution from these effects is found to be

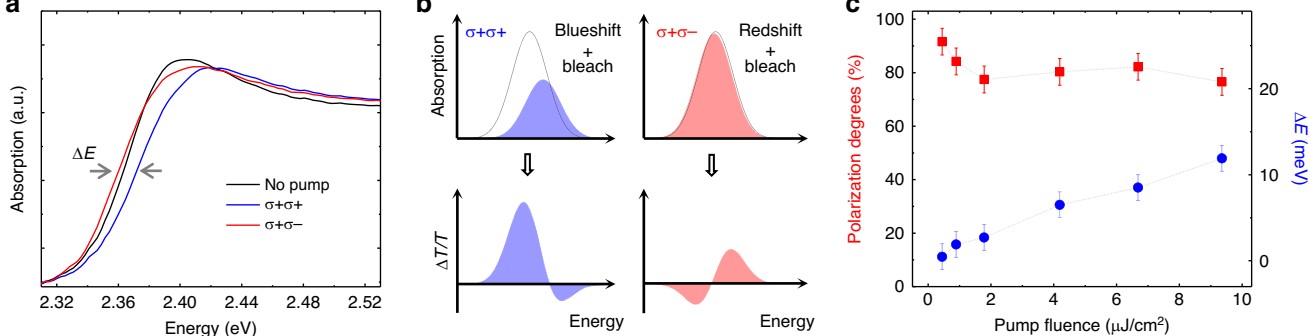

**Fig. 2 The photoinduced circular dichroism in $CsPbBr_3$ at room temperature. a** The absorption spectra without pump (black curve) and with pump at a delay time of 0.3 ps for σ+ σ+ (blue curve) and σ+ σ− (red curve) pump–probe configurations at room temperature. The horizontal gray arrows indicate energy shifts at the half maximum of absorption band edge, which define the energy splitting ($\Delta E$) between the blueshifted and redshifted absorption edges. The pump fluence is ~9.4 μJ/cm². **b** Schematics of pump-induced changes of the exciton resonance, mainly including the self-energy renormalization (blushift or redshift) and absorption bleaching, and the corresponding σ+ σ+ and σ+ σ− TA spectra. **c** Pump-fluence dependence of circular polarization degrees (red squares) and $\Delta E$ (blue circles) at 0.3 ps. The dashed lines are guide for the eye.

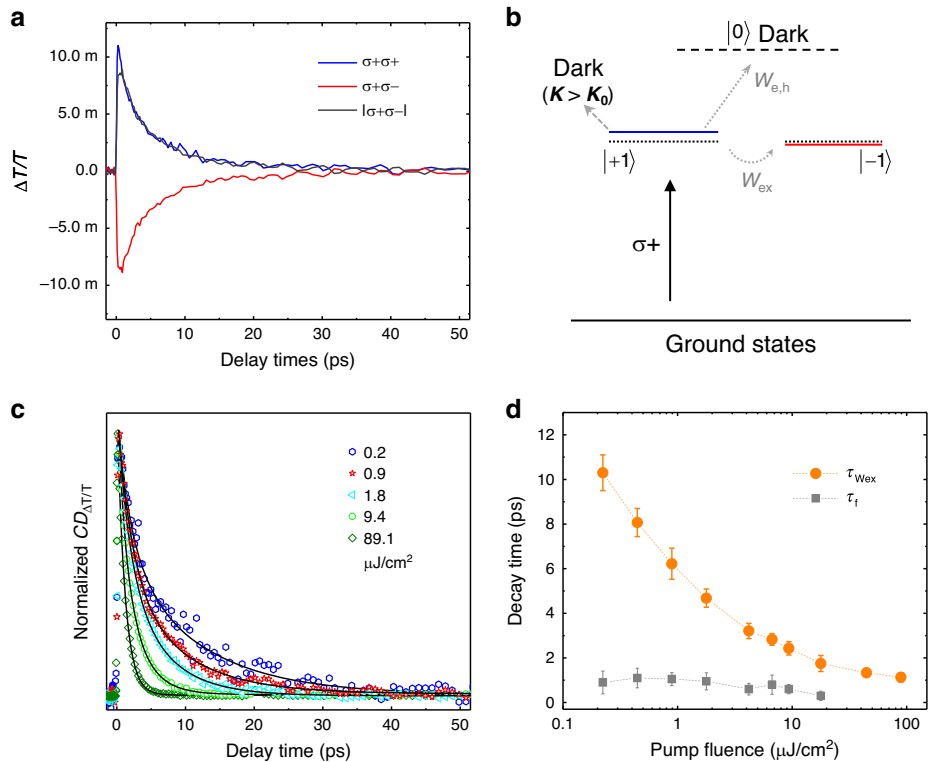

**Fig. 3 Exciton spin relaxation dominated by the electron–hole exchange interaction. a** The σ+ σ+ and σ+ σ− TA kinetics probed at ~2.35 eV, extracted from Fig. 1b, c, respectively. The gray curve shows the absolute value of the σ+ σ− TA kinetics. **b** The schematics of possible spin depolarization mechanisms for photo-injected +1 excitons. The $W_{e,h}$ represents the spin flip of electron or hole, and $W_{ex}$ is the electron–hole exchange interaction. $K$ and $K_O$ are the momentum of excitons and light, respectively. The horizontal dotted lines indicate the originally degenerate spin states. Following photoexcitation with a σ+ laser pulse, +1 exciton states shows a large blueshift (blue line) and −1 exciton states exhibits a slight redshift (red line). **c** Normalized kinetics of the circular dichroism signal $CD_{\Delta T/T}$, which is defined as $\Delta T/T_{\sigma+ \ \sigma+,\Delta t} - \Delta T/T_{\sigma+ \ \sigma-,\Delta t}$ and averaged in the probed energy range of 2.32–2.38 eV. The colored symbols are experimental results at different pump fluence, fitted with bi-exponential decay curves (black lines). **d** The spin decay times extracted from **c** as a function of pump fluence. The $\tau_{Wex}$ is the exciton spin decay times due to $e$–$h$ exchange interactions, while $\tau_f$ is the initial fast-decay component as discussed in the text. The dashed lines are guide for the eye.

not significant after examining the TA spectra at long delay times when the exciton spin fully relaxes and also those obtained with linearly polarized pump and probe pulses (see Supplementary Figs. 3 and 5).

**Exciton spin dynamics**. The spin-dependent optical non-linearities are linked to polarization degrees of excitons, thus allowing us to examine the exciton spin relaxation. In Fig. 3a, the TA signal of σ+ σ+ and σ+ σ− kinetics probed at ~2.35 eV decreases and increases as a function of delay time, respectively, and merge at ~50 ps. This indicates that the population of initially injected +1 excitons decreases, while the population of −1 excitons builds up gradually with increasing delay times. An equilibrium state is reached with an average population of +1 and −1 excitons in the system once the exciton spin fully relaxes after 50 ps. The decay/growth of +1/−1 exciton are symmetrical, except a minor fast decay of +1 excitons in the first few ps (Fig. 3a), providing key information for the spin relaxation mechanism. As shown in Fig. 3b, when the +1 excitons are injected through σ+ resonant excitations, major spin depolarization or decay channels include: (1) thermalization of excitons[46], (2) spin flip of either the electron or hole ($W_{e,h}$)[47,48], (3) spin flip of both constituents simultaneously via the electron–hole ($e$–$h$) exchange interaction ($W_{ex}$)[47,48], and (4) the recombination of excitons. The fourth one will be ignored here because it happens in the nanosecond timescale (see Supplementary Fig. 3). During the thermalization of excitons through exciton–exciton

and/or exciton–phonon scattering, partial exciton population will be scattered out of the so-called light cone and becomes momentum-dark[46]. Although the total population of +1 excitons does not change, this process can reduce the magnitude of absorption bleaching. Owing to strong carrier/exciton–phonon interaction in $CsPbBr_3$ at room temperature, the thermalization of excitons completes in a sub-ps timescale (see Supplementary Fig. 3). Furthermore, single fermion spin flip in +1 excitons leads to the population of spin-forbidden dark excitons ($J_z = 0$), but the $e$–$h$ exchange interaction flips +1 excitons into −1 excitons[48]. Therefore, the $e$–$h$ exchange interaction is the leading spin relaxation mechanism here supported by the symmetrical dynamics of +1 and −1 excitons[48].

Generally, there are three types of spin relaxation mechanisms in semiconductors known as Elliott–Yafet (EY), D'yakonov–Perel (DP), BAP mechanism[36]. For MHPs, spin depolarization mechanisms are complicated and strongly depend on sample quality[18], organic or inorganic cations[19,21], structural phases[14,49], dimensionality[15,16,50], lattice temperature[18,19,21], and so on. For instance, the EY mechanism through impurities or grain boundaries scattering plays a major role in solution-processed MHPs. Carrier spin relaxation lifetime is few ps in polycrystalline $CH_3NH_3PbI_3$ studied by using ultrafast pump–probe and time-resolved Faraday rotation (TRFR) spectroscopy[18]. Meanwhile, nanosecond exciton spin coherence was observed in polycrystalline $CH_3NH_3PbCl_xI_{3-x}$ at low temperatures by using TRFR[19,49]. This comes from an interplay between the Rashba effect and

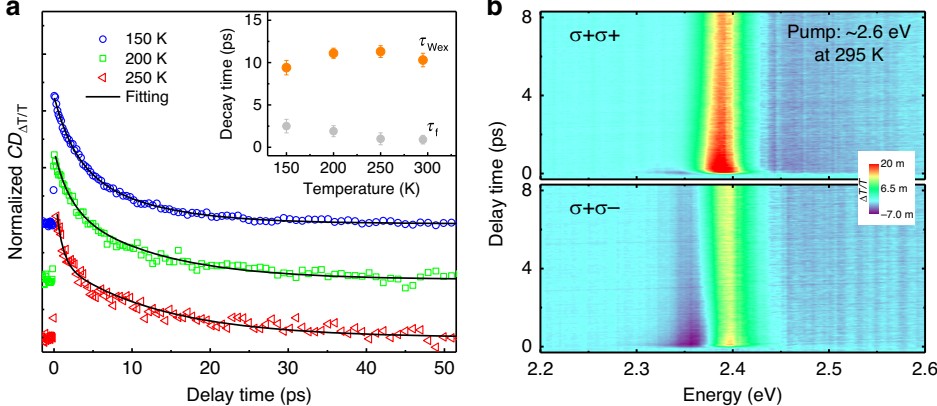

**Fig. 4 Temperature and pump-energy dependence of exciton spin dynamics. a** Normalized kinetics of $CD_{\Delta T/T}$ with vertical offsets as a function of lattice temperature at a fixed pump fluence of ~0.2 μJ/cm². The black lines are the fitting results with bi-exponential decays. The inset shows exciton spin decay times ($\tau_{Wex}$, orange circles) due to the exchange interaction and the initial fast-decay component ($\tau_f$, gray circles) at different temperature. **b** Time evolution of σ+ σ− and σ+ σ− TA spectra under a nonresonant pumping at room temperature. The pump fluence is ~0.9 μJ/cm².

distinct piezoelectric coupling, which overall contributes to a strongly suppressed DP mechanism[19,49]. Furthermore, the nanosecond spin coherence of localized electrons and holes was observed in solution-grown CsPbBr$_3$ crystals by using TRFR[21]. However, such spin coherence time for either excitons or carriers shortens quickly with increasing temperature and disappears at room temperature[19,21]. While strong excitonic exchange interactions (the BAP mechanism) leads to a short exciton spin lifetime of ~0.2 ps in two-dimensional (2D) MHPs[50], similar to other 2D semiconductors[51]. Here, we examine the exciton spin dynamics in high-quality CsPbBr$_3$ single-crystalline films. The EY mechanism is not important here because the impurities or grain boundaries are minimal. In addition, the large bandgap in CsPbBr$_3$ prevents a significant spin mixing[49]. The DP mechanism is also negligible since bulk CsPbBr$_3$ has inversion symmetry[14,15,49]. The exciton binding energy is about 40 meV in CsPbBr$_3$[12], providing an excellent balance of robust exciton resonances, optical nonlinearities and relatively long exciton spin lifetimes controlled by the BAP mechanism. The exciton spin lifetime of ~20 ps is so far the longest one observed in the MHP family at room temperature. In addition, the minor fast-decay component of the +1 excitons could arise from a combination of the thermalization effect and the single fermion spin flip.

The exciton spin dynamics also show a strong dependence on pump fluence. In Fig. 3c, the normalized CD signal ($CD_{\Delta T/T}$), defined as $\Delta T/T_{σ+ σ+} - \Delta T/T_{σ+ σ-}$, is plotted versus decay time for five different pump fluences. The decay dynamics of CD signals can be well-fitted with bi-exponential curves including a fast and a slow component at low and moderate pump fluence. The fitting results of spin decay times are summarized in Fig. 3d. The fast component, corresponding to the initial fast decay of +1 excitons (Fig. 3a), shows a sub-ps lifetime ($\tau_f$) and decreases slightly with increasing pump fluence. On the other hand, the dominant slow component has the longest decay time ($\tau_{Wex} \approx$ 10 ps) at low pump fluence (~0.2 μJ/cm²), corresponding to an exciton spin relaxation time of ~20 ps[52]. $\tau_{Wex}$ exhibits a pronounced decrease as a function of pump fluence (Fig. 3c, d). At high pump fluence (~90 μJ/cm²), the CD signal only shows a single-exponential decay time of ~1 ps. This short exciton spin lifetime indicates that the inter-exciton exchange interactions provide additional spin relaxation channels at high exciton densities, primarily through the $e$–$e$ and $h$–$h$ spin–spin couplings among excitons in line with those proposed in GaAs quantum wells[34,53]. The excitonic Mott transition from excitons to the $e$–$h$ plasma plays a negligible role at low and medium pump fluence

because the screening effect from excitons are very weak[45]. At sufficiently high pump fluence when the $e$–$h$ plasma prevails, significant bandgap renormalizations and fast spin relaxation are still expected due to spin-sensitive interactions among electrons and holes[34,38]. The intriguing phenomena presented here require sophisticated theoretical calculations in order to be further deciphered in a quantitative way.

Figure 4a presents the decay dynamics of CD signals at three different temperatures with a low excitation density. The exciton spin decay time does not show noticeable temperature dependence (inset in Fig. 4a and Supplementary Fig. 6), another characteristic of the BAP mechanism, because the Coulomb interaction within excitons stays constant with varying lattice temperatures[54]. In contrast, the spin dynamics of carriers or excitons controlled by the EY or DP mechanism show strong temperature dependences in previous work on MHPs[18,19,21]. Meanwhile, the fast-decay component shows a slightly increase with decreasing lattice temperature since both the thermalization effect and the single fermion spin flip happen on a longer timescale at low temperature[18,19,21,46]. Furthermore, we observe that the spin-dependent optical nonlinearities are quite robust even with photoexcitation well above its bandgap. In Fig. 4b, the σ+ σ+ and σ+ σ− TA spectra obtained with a pump beam at ~2.6 eV have similar but less prominent features compared with those in Fig. 1b, c, respectively. Following non-resonant excitation, spin-polarized hot carriers are injected into the system and subsequently relax down to the band edge mostly through efficient carrier-phonon and carrier–carrier scattering[40]. Therefore, a portion of electrons and holes loses their spin orientation before forming excitons[36]. Indeed, a circular polarization degree of ~30% is seen at ~1.5 ps at which the exciton population is fully built up, but the spin relaxation time of excitons is the same as those with resonant excitations (see Supplementary Fig. 7). The intriguing spin-contrasting phenomena are observed in single-crystalline CsPbCl$_3$ and CsPbBr$_x$Cl$_{3−x}$ at room temperature (see Supplementary Fig. 8). The exciton spin lifetime in CsPbCl$_3$ is slightly shorter than that in CsPbBr$_3$ under the same pump fluence. Since the exciton binding energy in CsPbCl$_3$ is ~70 meV[13], the $e$–$h$ exchange interaction within the exciton is stronger than that in CsPbBr$_3$, and therefore a shorter exciton spin lifetime is expected. Lastly, the time- and polarization-resolved photoluminescence (PL) decay measurements unambiguously reveal the spin dynamics of +1 and −1 excitons under a σ+ excitation at 3.1 eV in CsPbBr$_x$Cl$_{3−x}$ at room temperature (see Supplementary Figs. 9 and 10). The degree of circularly

polarized emission of excitons reaches a maximum of ~20% at ~16 ps after photoexcitation (limited by the temporal resolution of the setup) and subsequently decays on a longer timescale. However, the actual exciton spin lifetime cannot be extracted in such PL spectroscopy measurements due to the insufficient temporal resolution of the setup (see "Methods" and Supplementary Figs. 9 and 10).

## Discussion

In summary, we have studied spin-contrasting optical non-linearities in single-crystal $CsPbX_3$ by using ultrafast TA spectroscopy. Upon the injection of $+1$ excitons into the system, the energy degeneracy of two spin states is lifted, leading to a photoinduced CD. The energy splitting between $+1$ and $-1$ exciton states is attributed to a strong repulsion for excitons with parallel spins and a weak attraction between excitons with anti-parallel spins. The energy splitting is ~12 meV at a moderate pump fluence of ~9.4 μJ/cm$^2$ and can further increase to several tens of meV at high pump fluences. Although similar phenomenon can be achieved by applying magnetic fields, such giant spin splitting is beyond the reach with actual magnetic fields in the laboratory[21]. Surprisingly, the spin-contrasting optical nonlinearities of excitons in bulk $CsPbX_3$ are more prominent than those in 2D semiconductors such as 2D hybrid organic–inorganic perovskites[50], monolayer transition metal dichalcogenides[55,56], and GaAs-based quantum wells[32,33,41] in terms of the magnitude of the spin splitting. Furthermore, the dominant exciton spin relaxation mechanism is revealed to be the intra- and inter-exciton exchange interaction at low and high excitation densities, respectively. The exciton spin relaxation time can be as long as ~20 ps at low excitation densities, which is longer than the lifetime of excitonic polaritons (~3 ps)[12], and thus spin-polaritons is expected to form at room temperature in $CsPbX_3$. With a fascinating combination of high spin injection rate, strong optical nonlinearities and long exciton spin lifetimes at room temperature, $CsPbX_3$ hold great promise for future optoelectronics and quantum-optics applications with novel spin functionalities[26–28].

## Methods

**Sample growth.** The $CsPbX_3$ perovskite films are synthesized by a vapor transport method using a home-built CVD system[5,13,31]. The growing mechanism is based on Van der Waals epitaxy, which is well-established to produce high-quality single crystals[5,13,31]. Fresh muscovite mica is exfoliated as substrates and placed inside the downstream of a quartz tube mounted in a single zone furnace (Lindberg/Blue MTF55035C-1). A quartz boat with 0.15 g $CsPbBr_3$ powders are put inside the middle of quartz tube. The quartz tube is pumped down to a base pressure of 45 mTorr, which is followed by a 30 sccm flow of high-quality $N_2$ (99.999%). The pressure and temperature for the synthesis are set to 40 Torr and 600 °C, respectively. The whole synthesis time is set to 30 mins. The thickness of the synthesized $CsPbBr_3$ thin films can be controlled from 60 nm to several micrometers.

The source material for the synthesis are obtained through a solution method. 0.3 mmol CsBr and 0.3 mmol $PbBr_2$ are dissolved in 5 ml $H_2O$ and 5 ml hydrobromic acid (48%, Sigma-Aldrich), respectively. The CsBr and $PbBr_2$ solutions are mixed and stirred for half an hour. The $CsPbBr_3$ precipitates are washed in ethanol for 5 times and dried in an oven.

**Sample characterization.** The optical and fluorescence images are obtained by using an Olympus BX51 fluorescence microscope. The AFM measurements are conducted with a commercial setup (Cypher ES, Oxford Instruments) by using the tapping mode. The SEM images are collected on a field-emission scanning electron microscope (FE-SEM, JEOL JSM-7001F) operated at 5 kV. The high-resolution XRD was performed on Bruker D8 Discover Diffractometer with Cu-Kα radiation ($\lambda = 1.5406$ Å).

**Ultrafast TA spectroscopy.** The ultrafast pump-probe measurements are conducted by employing a commercial TA spectrometer (Ultrafast Systems, USA). The 800 nm output pulse laser (1 kHz repetition rate, ~100 fs pulse width) from a Ti: sapphire regenerative amplifier (Spectra-Physics Spitfire) is split into two paths. One beam goes through a mechanical delay stage to pump a $CaF_2$ or sapphire crystal to generate a light continuum to serve as the probe pulse. The second beam is sent to an optical parametric amplifier (Spectra-Physics TOPAS) to generate

pump pulses (260–2000 nm). The circularly polarized pump and probe beams are obtained by using quarter-wave plates purchased from Thorlabs. The pump and probe pulses are non-collinearly focused on samples with a beam size of about 100 and 50 μm, respectively, by using parabolic mirrors. A mechanical chopper with a synchronized readout of a CMOS detector is used for acquisitions of probe spectra with and without pump-induced changes, enabling the measurement of a differential transmission. The spectra resolution is ~1.0 nm across the detecting range. For each TA spectrum, three scans are performed to ensure the repeatability for the obtained results. The steady-state and ultrafast optical spectra at low temperatures are conducted by using a commercial cryostat (Cryo Industries of America, INC.).

**Optical characterization.** The absorption spectra of $CsPbX_3$ thin films are measured using a PerkinElmer Lambda 950 UV–vis–IR spectrometer. The PL spectra of $CsPbX_3$ are performed with a home-built confocal micro-PL setup. All measurements are carried out at room temperature. A Ti:sapphire femtosecond-pulsed laser with ~100 fs pulses at 80 MHz is used as the excitation source. The emission of the laser is frequency-doubled by using a BBO nonlinear crystal to output 400 nm pulses and is focused (50× objective lens, NA = 0.65) onto the sample. The laser spot size at focus is about 1 μm in width. The laser power is kept at ~1.2 μW.

The PL emission, which is separated from the excitation laser by a combination of a 90:10 beam splitter and a long-pass filter, is dispersed in a 320 mm spectrometer together with a 1800 lines/mm grating. For the time-averaged measurements, the PL emission is detected using a liquid nitrogen-cooled Si charge coupled device detector. For time-resolved decay measurements, the PL emission is first spectrally filtered with the spectrometer, and the filtered PL emission is detected with an avalanche photon detector connected to a single photon counting module (PicoHarp 300). The overall temporal resolution of the time-resolved setup in the visible wavelength range is about 40 ps. For polarization-resolved measurements, a quarter-wave plate from Thorlabs is placed before the objective to generate the circularly polarized excitation and converts the collected emission into two orthogonal linear polarization components. A motorized half-wave plate and a linear polarizer are placed before the entrance slit of the spectrometer. The half-wave plate is rotated, while the linear polarizer is fixed to ensure that any change in the PL intensity is due to the intrinsic polarization of the emission and not due to the polarization response of the spectrometer grating.

## Data availability

The data that support the findings of this study are available from the corresponding author upon request.

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

## Acknowledgements

Q.X. gratefully acknowledges the funding support from Singapore Ministry of Education via AcRF Tier 3 Program "Geometrical Quantum Materials" (MOE2018-T3-1-002), AcRF Tier 2 grants (MOE2018-T2-1-040 and MOE2019-T2-1-004), and Tier 1 grants (RG 194/17). Y.H. acknowledges financial support from the Knut and Alice Wallenberg Foundation.

## Author contributions

W.Z. and Q.X. conceived the idea and designed the research. W.Z. performed the ultrafast transient absorption spectroscopy measurements. R.S. and J.W. prepared the cesium lead halides perovskites films and conducted measurements of optical microscopy, scanning electron microscope (SEM), and atomic force spectroscopy (AFM). J.F. conducted the X-ray diffraction (XRD) measurements. Y.H., W.Z., and C.F.F. performed the time-resolved photoluminescence measurements. W.Z. and Q.X. analyzed the data and wrote the paper. All authors commented on the paper.

## Competing interests

The authors declare no competing interests.

## Additional information

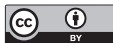

