## [Peer Review File · Nature Communications]

REVIEWER COMMENTS

Reviewer #1 (Remarks to the Author):

This manuscript reports on the TA circular dichroism and spin lifetime in CsPbBr₃ films. The measurements and analysis are fairly interesting and worth publication. There have been other such studies in the literature but the authors here report that for these samples the spin lifetimes at RT can be 20 ps at low carrier densities. They report on interesting temperature dependence and carrier density dependence.

I have a few points that I think the authors should address prior to publication.

(1) The samples are not well described. In some cases the authors state that they are single crystals - but this is not shown or discussed. In the SI and methods section there is some more information. They claim single crystal films?? But also describe the methods for preparing powders. What's the thickness of the films? What's the crystallinity?

(2) It looks like from fig. 4 that there is no temperature dependence of the longer time constant. Maybe I missed it - but is there a temperature dependence of the early time dynamics (the fast component). It looks like it from the figure. What is the reason for this. The authors mention this but do not discuss.

Reviewer #2 (Remarks to the Author):

Zhao et al report the observation of the photoinduced circular dichroism in the single crystal form of CsPbX₃ thin film using time- and polarized-resolved photo-induced absorption spectroscopy. Their discovery is two-fold: (1) they found a pronounced spin splitting up to 23 meV between the 'spin-up' and 'spin-down' exciton states under the right-handed ($\sigma+$) circularly polarized excitation, which acts like a pseudo-magnetic field to lift the spin degeneracy at both low and high pump fluences; (2) They observed a remarkable long exciton spin relaxation time (~ 20 ps) at room temperature at the low pump fluence. The manuscript is well-structured and written. The proposed physical picture has been conveyed through the entire manuscript. This research is timely and will be impactful for the newly launched field of perovskite spintronics. While the pump-probe technique here (including ultrafast magneto-optical Kerr effect) is not novel and have been traditionally employed for understanding the spin and photo-physics in inorganic semiconductors, the application of time- and polarized-resolved photo-induced absorption spectroscopy for excitonic spin dynamics in this emergent solution-process single crystal thin film is unique and not trivial. Many researchers in the field of hybrid/inorganic perovskites optoelectronics emphasize more device performance and material development but lack this capability to fully characterize the exotic spin dynamics and their spin functionalities, whereas many researchers who possess this technique focuses more on the field of metallic/semiconductor spintronics. This work bridges the gap between ultrafast spintronics and solution-processed perovskite materials which are still at its nascent stage caused by the misleading concept that the reduced-order via cruder solution processing would diminish the classical spin relaxation model, e.g., Bir-Aronov-Pikus (BAP) as demonstrated in this work. Beyond the rise of hybrid organic-inorganic perovskites (e.g., MAPbBr₃, 2D layered perovskites, etc.) for spintronic studies, adding the inorganic metal halide material class with excellent environmental stability and robust excitonic properties would substantially enrich the family of solution-processed hybrid materials for future opto-spintronics and spin-optoelectronic applications. Therefore, I would further consider the publication of this manuscript after my following concerns are addressed.

[1] The observation of the photoinduced circular dichroism is indeed undoubtful, corroborated by the carefully designed control experiments and measurement protocols. The direct correlation between the observed circular dichroism and pseudo-magnetic field from the circularly polarized excitation, however, is relatively weak. This concern may be caused by a lack of similar observation under the Left-handed ($\sigma-$) polarized excitation. It would be anticipated that the generated pseudo-magnetic field will be opposite under the different polarized excitation, by which

the separation of the spin up and spin down exciton states should be reversed, as well as the circular polarization. I have checked the method section: the different circular polarization of the pump could be generated through the optics from Thorlabs. Moreover, have authors performed a similar experiment by flipping the sample to exclude the potential contributions from the substrate? I would ask the authors to comment on the missing of these key results in the manuscript.

[2] I appreciate the clear physics picture presented in the manuscript, but I am still not fully convinced by the proposed 'nonlinearity' that has been emphasized several times in the manuscript. In Fig. 2c, the energy splitting caused by the pump increases when the pump fluence increases, following an approximately linear response. On page 3, the first paragraph, it says: "...The photoinduced spin splitting is ~ 12 meV at a moderate excitation density (~ 9.4 uJ/cm²).". In the caption of S.I. Fig. S3, it shows that " ΔE can be ~ 23.0 meV at a pump fluence of 17.8 uJ/cm²". We could find the splitting caused by the pump power does follow a roughly linear response.

[3] I have a concern about the inconsistency between the proposed linearly increased ΔE caused by the high pump fluence and a roughly constant value of the circular polarization as shown in Fig. 2c. According to the proposed model, a higher pump fluence induces a larger energy gap between two exciton spin states (A splitting of 23 meV at 17.8 uJ/cm² is comparable to the thermal fluctuation at the room temperature), from which a high circular polarization/spin polarization would be envisioned. Why the slightly reduced circular polarization is observed?

In Fig. 3c, the authors used the second definition about the circular polarization degree (in the form of CD signal). The CD signal monotonically decays when the pump fluence increases as well as the derived 'spin lifetime' as shown in Fig. 3d. When the energy gap induced by the high pump fluence is significantly increased, it would be difficult for flipping spin and transition between two spin states because of the larger energy barrier. It is puzzled to me that e-h exchange interactions are not suppressed. Or there would be two mechanisms that compete with each other that eventually produce a roughly constant value of polarization (80%-90%) as presented in Fig. 2c?

[4] I am not fully convinced about the blue/redshift observed in the TA spectroscopy can be attributed to the spin splitting state unless the spin characters of the absorption peak are confirmed, particularly this also occurs at a low pump fluence (0.9 uJ/cm²). Will this be caused by the photoinduced birefringence in the CsPbX₃ single crystal which also produced a pseudo-magnetic field signal? Have the authors conducted the angular dependent circular polarization by rotating the sample along the in-plane direction?

An optical Hanle effect (i.e., spin precession under the vertical applied magnetic field, Ref. 17 and 26) would be suitable 'smoking-gun' tool to unambiguously determine the spin characters of the exciton states and their spin relaxation times at different pump fluences. By applying the magnetic field, the precessed spin mixes two spin-polarized exciton states. Thus the field-dependent circular polarization degree will be observed. The circular polarization signal with the σ^+ probe will decrease while the signal with the σ^- probe will increase when the applied external magnetic field increases. The spin relaxation time could be precisely derived from the HWFM of the curve.

I would also suggest the authors check the time-resolved circular polarization of the photoluminescence response which directly provides the experimental proof the spin characters and the radiative decays of the spin-dependent excitons as demonstrated in Ref. 20.

Minor issues:

[5] The basic characterizations of the single crystal form of the prepared thin film are missing in the S.I., such as XRD, roughness and surface morphology by the AFM (not the only the 'profilometer'), particularly the circular dichroism (CD) spectra for the thin film without the pump.

[6] In Fig. 3c, the olive square represents the decay at 89.1 uJ/cm^2 . But there are only olive diamonds shown in the figure. The y-axis would be better in the unit of circularly polarization degree.

[7] In Fig. 3b, the lifted spin degeneracy for the spin up and spin down exciton states induced by the circularly polarized pump is not presented.

Reviewer #3 (Remarks to the Author):

In this work the spin dynamics of photoexcitations in the popular inorganic perovskite semiconductors, namely CsPbBr₃ has been studied using the transient circularly polarized pump probe technique at various excitation intensities and temperatures. The experimental part is not new in the field. Spin dynamics using the same technique has been studied in various 2D and 3D hybrid organic inorganic perovskites, including by the authors. Thus there is no novelty here.

In addition the message of this work is not proven unambiguously. Firstly there is no experimental proof that the circular polarized bands that are separated by the giant energy of few tens of meV belong to the same state to begin with. In addition, authors mention many-body interaction as a mechanism for the giant photoinduced circular dichroism, but have not done any calculation to show it. This is absolutely needed for publication in high profile journal such as Nature Commun. Authors mention other phenomena such as bandgap renormalization and state filling, but have not tried to use these mechanisms to fit the obtained polarized spectra. Thus the interpretation of the results falls in the realm of speculation.

Also authors refer several times papers on GaAs MQW from the 90th, where the relation between the exciton and band-edge carriers was not recognized. In the meantime the inter-relation between excitons and the continuum in the absorption spectrum of a semiconductor, especially for relatively small exciton binding energy such as in CsPbBr₃ has been advanced. Authors have not taken this into account in their discussion.

For these reasons I do not recommend publication of this work in Nature Commun.

Point-to-Point Response to Reviewer's Comments:

Reviewer #1:

This manuscript reports on the TA circular dichroism and spin lifetime in CsPbBr₃ films. The measurements and analysis are fairly interesting and worth publication. There have been other such studies in the literature but the authors here report that for these samples the spin lifetimes at RT can be 20 ps at low carrier densities. They are report on interesting temperature dependence and carrier density dependence.

Response: We greatly appreciate the Reviewer for the positive feedback and his/her recommendation for publication of our work. The Reviewer's constructive comments are very helpful for us to further improve the quality of this manuscript. We have carefully considered all the Reviewer's comments and revised the manuscript accordingly with additional experimental results and data analysis.

I have a few points that I think the authors should address prior to publication.

(1) The samples are not well described. In some cases the authors state that they are single crystals - but this is not shown or discussed. In the SI and methods section there is some more information. They claim single crystal films?? But also describe the methods for preparing powders. What's the thickness of the films? What's the crystallinity?

Response: We thank the Reviewer for these critical comments and suggestions. We have conducted necessary characterizations to confirm the single-crystal nature of our samples. In the revised Supplementary Fig.1, the fluorescence, AFM and SEM images and high-resolution XRD of CsPbBr₃ thin films are presented. We also present the transmission electron microscopy (TEM) characterization of CsPbBr₃ films transferred onto the copper grid as shown below for review only since this is still an ongoing collaborating project for a different topic. All these characterizations help to justify that CsPbBr₃ thin films are indeed high-quality single crystals. In addition, we also cited our previous work which elaborated in detail the synthesis and characterizations (Ref. 5, 13 and 31).

The above figure shows a selected area electron diffraction (SAED) pattern of the CsPbBr₃

thin film along the [001] zone axis. The CsPbBr₃ sample on the copper grid is highlighted by the dashed circle which has a diameter of ~60 microns. The TEM characterizations were conducted using the JEOL JEM-ARM200F atomic resolution analytical electron microscope operating at an accelerating voltage of 200 KV.

The above schematics shows the synthesis setup using a home-built vapor-transport system. The CsPbBr₃ powder was used as source materials to grow the single-crystal thin films. In the Methods section, we have added more detailed information for the Sample Growth, especially the method to prepare the source powders. Our first vapor transport synthesis of hybrid perovskite films and all inorganic perovskite crystals were reported in our previous work (Ref. 5, 13 and 31).

The CsPbBr₃ thin films with the thickness of ~200 nm were used to obtain the transient absorption data presented in the main text, as revealed by AFM measurements (Fig. S1). Moreover, we have checked more CsPbBr₃ samples with different thickness (100~300 nm) and did not observe any notable thickness dependence for the transient spin-dependent optical phenomena, i.e., the long spin relaxation time and large energy splitting of excitons.

(2) It looks like from fig. 4 that there is no temperature dependence of the longer time constant. Maybe I missed it - but is there a temperature dependence of the early time dynamics (the fast component). It looks like it from the figure. What is the reason for this. The authors mention this but do not discuss.

Response: We thank the Reviewer for the constructive comments. The lifetimes of the fast-decay component at different temperatures are presented in the revised Fig. 4a following the Reviewer's suggestion. The decay time of fast-decay component slightly increases with the decrease of lattice temperatures. As discussed in the revised main text (Paragraph 2 at Page 7), the fast-decay component is believed to be resulted from a combination of the thermalization of excitons and the single fermion spin flip through either the EY or DP mechanism. When the lattice temperature decreases, the thermalization of photo-injected excitons would happen in a longer timescale because the phonon population and the subsequent exciton-phonon scattering decrease with temperature [Ref.46]. On the other hand, it has been well demonstrated that the spin relaxation time of excitons/carriers due to the EY or DP mechanism increase with the decrease of lattice temperature in metal halide perovskites [Ref. 18, 19 and 21]. Therefore, both the thermalization effect and the single fermion spin flip

can lead to a longer decay time of the fast-decay component at low temperature in agreement with our experimental results.

Reviewer #2:

Zhao et al report the observation of the photoinduced circular dichroism in the single crystal form of CsPbX₃ thin film using time- and polarized-resolved photo-induced absorption spectroscopy. Their discovery is two-fold: (1) they found a pronounced spin splitting up to 23 meV between the 'spin-up' and 'spin-down' exciton states under the right-handed (σ^+) circularly polarized excitation, which acts like a pseudo-magnetic field to lift the spin degeneracy at both low and high pump fluences; (2) They observed a remarkable long exciton spin relaxation time (~ 20 ps) at room temperature at the low pump fluence. The manuscript is well-structured and written. The proposed physical picture has been conveyed through the entire manuscript. This research is timely and will be impactful for the newly launched field of perovskite spintronics. While the pump-probe technique here (including ultrafast magneto-optical Kerr effect) is not novel and have been traditionally employed for understanding the spin and photo-physics in inorganic semiconductors, the application of time- and polarized-resolved photo-induced absorption spectroscopy for excitonic spin dynamics in this emergent solution-process single crystal thin film is unique and not trivial. Many researchers in the field of hybrid/inorganic perovskites optoelectronics emphasize more device performance and material development but lack this capability to fully characterize the exotic spin dynamics and their spin functionalities, whereas many researchers who possess this technique focuses more on the field of metallic/semiconductor spintronics. This work bridges the gap between ultrafast spintronics and solution-processed perovskite materials which are still at its nascent stage caused by the misleading concept that the reduced-order via cruder solution processing would diminish the classical spin relaxation model, e.g., Bir-Aronov-Pikus (BAP) as demonstrated in this work. Beyond the rise of hybrid organic-inorganic perovskites (e.g., MAPbBr₃, 2D layered perovskites, etc.) for spintronic studies, adding the inorganic metal halide material class with excellent environmental stability and robust excitonic properties would substantially enrich the family of solution-processed hybrid materials for future opto-spintronics and spin-optoelectronic applications. Therefore, I would further consider the publication of this manuscript after my following concerns are addressed.

Response: We greatly appreciate the Reviewer's recognition on the novelty and significance of our work. We carefully considered all the constructive suggestions and comments from the Reviewer and provided supplementary experimental results and data analysis accordingly. We

believe that all the concerns raised by the Reviewer are addressed thoroughly and the quality of the manuscript has been improved significantly.

[1] The observation of the photoinduced circular dichroism is indeed undoubtful, corroborated by the carefully designed control experiments and measurement protocols. The direct correlation between the observed circular dichroism and pseudo-magnetic field from the circularly polarized excitation, however, is relatively weak. This concern may be caused by a lack of similar observation under the Left-handed (σ^-) polarized excitation. It would be anticipated that the generated pseudo-magnetic field will be opposite under the different polarized excitation, by which the separation of the spin up and spin down exciton states should be reversed, as well as the circular polarization. I have checked the method section: the different circular polarization of the pump could be generated through the optics from Thorlabs. Moreover, have authors performed a similar experiment by flipping the sample to exclude the potential contributions from the substrate? I would ask the authors to comment on the missing of these key results in the manuscript.

Response: We thank the Reviewer for these constructive suggestions. Control experiments of the transient absorption (TA) spectroscopy are conducted with Left-handed (σ^-) polarized excitation and with samples on different substrates. From the σ^- - σ^- and σ^- - σ^+ TA spectra presented below, the -1 exciton states show a blueshift and +1 exciton states have a redshift when the -1 excitons are injected in the system. The opposite effect demonstrated with Left- and Right-handed polarized excitations indicates that the many-body interaction of spin-polarized excitons can function as an effective magnetic field.

Supplementary Fig. S2 shows time evolution of TA spectra of CsPbBr₃ obtained with σ^- - σ^- and σ^- - σ^+ pump-probe configuration, respectively, at room temperature.

We have checked the TA spectra by flipping the sample and found that the spin-dependent optical phenomena do not show any observable differences. Furthermore, we have transferred CsPbBr₃ thin films on other substrates like sapphire or TEM grid. We are confident that there is no residue of Mica after the transfer process by carefully performing TEM characterizations of the samples transferred on copper grids [for example the one discussed in the response of

Comments (1) raised by Reviewer1]. The TA spectra of CsPbBr₃ on sapphire shown below help us to conclude that the substrate effect is minimal. We would like to also emphasize that the transferring of perovskite thin films from mica growth substrate onto silicon or distributed Bragg reflector (DBR) substrate have been well documented by our group (Ref. 7, 12, and 25), which can even support microfabrication on those films. Therefore, there is no need of worrying about the mica residue or other related substrate issues.

Supplementary Fig. S4 shows time evolution of TA spectra of CsPbBr₃ on sapphire substrate obtained with the $\sigma+\sigma+$ (a) and $\sigma+\sigma-$ (b) pump-probe configuration, respectively, at room temperature. (c) The exemplary optical image of CsPbBr₃ thin films transferred onto double-polished sapphire substrate is shown.

[2] I appreciate the clear physics picture presented in the manuscript, but I am still not fully convinced by the proposed ‘nonlinearity’ that has been emphasized several times in the manuscript. In Fig. 2c, the energy splitting caused by the pump increases when the pump fluence increases, following an approximately linear response. On page 3, the first paragraph, it says: “...The photoinduced spin splitting is ~ 12 meV at a moderate excitation density (~ 9.4 uJ/cm²).” In the caption of S.I. Fig. S3, it shows that “ ΔE can be ~ 23.0 meV at a pump fluence of 17.8 uJ/cm²”. We could find the splitting caused by the pump power does follow a roughly linear response.

Response: We thank the Reviewer for the comment. The transient absorption spectroscopy is a non-linear optical method, in contrast to the linear or steady-state absorption spectroscopy. That is the reason we use “nonlinearity” in the manuscript. Therefore, the photoinduced spin splitting itself, rather than its magnitude against the pump fluence, is considered as a nonlinear optical phenomenon.

[3] I have a concern about the inconsistency between the proposed linearly increased ΔE caused by the high pump fluence and a roughly constant value of the circular polarization as shown in Fig. 2c. According to the proposed model, a higher pump fluence induces a larger energy gap between two exciton spin states (A splitting of 23 meV at 17.8 uJ/cm² is comparable to the thermal fluctuation at the room temperature), from which a high

circular polarization/spin polarization would be envisioned. Why the slightly reduced circular polarization is observed?

In Fig. 3c, the authors used the second definition about the circular polarization degree (in the form of CD signal). The CD signal monotonically decays when the pump fluence increases as well as the derived 'spin lifetime' as shown in Fig. 3d. When the energy gap induced by the high pump fluence is significantly increased, it would be difficult for flipping spin and transition between two spin states because of the larger energy barrier. It is puzzled to me that e-h exchange interactions are not suppressed. Or there would be two mechanisms that compete with each other that eventually produce a roughly constant value of polarization (80%-90%) as presented in Fig. 2c?

Response: We thank the Reviewer for the constructive comments. We apologize that the critical information may not be clarified properly which caused these confusions. The circular polarization as a function of pump fluence shown in Fig. 2c reflects the spin injection efficiency in the sample under optical excitation. When the Right-handed (σ^+) polarized laser pulse is used as excitation, 100% of +1 exciton would be injected in the sample ideally. However, there are possible optical processes which could decrease the spin injection efficiency. Due to the large spin-orbital coupling in metal halide perovskites, the spin injection efficiency is expected to be high [Ref.18]. Therefore, we believe that the carrier-carrier (or exciton-exciton), carrier-phonon and/or carrier-impurity scattering, happened during the polarization-population conversion process at the early delay time ($< \sim 0.3$ ps), are responsible for the slight spin depolarization of excitons (10-20%) after photoexcitation. With the increase of pump fluence, carrier-carrier scattering is enhanced and thus leads to a further decrease of the spin injection efficiency as shown in Fig. 2c in the main text.

When the population of excitons builds up completely at ~ 0.3 ps, the spin splitting arising from excitonic many-body interaction reaches its maximum and subsequently decays following the spin relaxation of excitons at longer delay times. If such a large energy splitting between two spin states is induced by the magnetic field, we would expect the spin lifetime to be much longer as elaborated by the Reviewer. Here, the spin splitting is induced by the many-body interaction among spin-polarized excitons. A stark trade-off between spin splitting and spin lifetime is imposed. When a high density of excitons is injected in the system at high pump fluence, the spin splitting is initially large (at ~ 0.3 ps). On the other hand, because the excitons are very close to each other, an additional spin-depolarization channel via the inter-exciton e-h exchange interaction is introduced and leads to shorter exciton spin lifetimes (Fig. 3c and 3d). The CD signal shown in Fig. 3c are positively correlated to the circular polarization degree and thus were fitted with exponential decays to extract the exciton spin decay time.

[4] I am not fully convinced about the blue/redshift observed in the TA spectroscopy can be attributed to the spin splitting state unless the spin characters of the absorption peak are confirmed, particularly this also occurs at a low pump fluence (0.9 uJ/cm^2). Will this be caused by the photoinduced birefringence in the CsPbX_3 single crystal which also produced a pseudo-magnetic field signal? Have the authors conducted the angular dependent circular polarization by rotating the sample along the in-plane direction?

An optical Hanle effect (i.e., spin precession under the vertical applied magnetic field, Ref. 17 and 26) would be suitable 'smoking-gun' tool to unambiguously determine the spin characters of the exciton states and their spin relaxation times at different pump fluences. By applying the magnetic field, the precessed spin mixes two spin-polarized exciton states. Thus the field-dependent circular polarization degree will be observed. The circular polarization signal with the σ^+ probe will decrease while the signal with the σ^- probe will increase when the applied external magnetic field increases. The spin relaxation time could be precisely derived from the HWFM of the curve.

I would also suggest the authors check the time-resolved circular polarization of the photoluminescence response which directly provides the experimental proof the spin characters and the radiative decays of the spin-dependent excitons as demonstrated in Ref. 20.

Response: We thank the reviewer for the constructive suggestions and comments. The CsPbBr_3 thin films exhibit orthorhombic structure at room temperature evidenced by the high-resolution XRD data shown in Fig. S1f, which is a birefringent structure. We have conducted the polarization dependent photoluminescence (PL) measurements and did not find any energy splitting which is linked to the birefringence in our samples. Fig. S1e in the revised Supplementary Information shows the experimental results and detailed discussions.

The Hanle effect was conducted with MAPbBr_3 at 10 K in Ref. 26 [Wang J. et al., Nat. Commun. 10, 129 (2019)] in the previous version of the manuscript (which is Ref. 28 in the current version). Because a weak circular polarization ($\sim 3\%$) in the PL spectra of MAPbBr_3 can be observed at zero magnetic field using circularly polarized *cw* excitation, the authors were able to further measure circular polarization as a function of the magnetic field and extract the effective exciton spin lifetime. However, we could not observe circular polarizations in the steady-state PL spectra of CsPbX_3 at the temperature range checked. This is because the exciton spin lifetime is much shorter than the exciton radiative decay time. Therefore, the Hanle effect could not be measured with CsPbX_3 thin films.

Following the Reviewer's suggestion, we conducted the time-resolved photoluminescence (TRPL) decay measurement with a home-built time-correlated single-photon counting (TCSPC) setup. The PL emission is first spectrally filtered with the spectrometer, and then the filtered PL

emission is detected with an avalanche photon detector (APD) connected to a single photon counting module (PicoHarp 300). The overall temporal resolution of the setup is ~ 40 ps in the visible range. The TRPL results of CsPbX_3 samples are shown below (i.e., Fig.S9 and Fig.S10) with a pulsed laser excitation at 3.1 eV (see Methods). The exciton spin dynamics of $\text{CsPbBr}_x\text{Cl}_{3-x}$ thin films can be detected by the TRPL measurements. However, the actual exciton spin lifetime is difficult to be quantitatively extracted limited by the temporal resolution of the setup (~ 40 ps).

Supplementary Fig. S9. (a) Room-temperature PL (red curve) and absorbance (blue curve) spectra of the $\text{CsPbBr}_x\text{Cl}_{3-x}$ thin film. The large emission Stokes (~ 80 meV) shift is not due to defects or impurities but from the lattice anharmonicity in carrier-lattice interactions which is intrinsic to metal halide perovskites [Guo Y. et al., *Nat. Commun.* 10, 1175 (2019)]. The three black arrows indicate the emission energy which are selected to check the PL decay. (b) The PL decay curves with the $\sigma+\sigma+$ (blue curve) and $\sigma+\sigma-$ (red curve) excitation-detection polarization configuration at ~ 2.6 eV. (c) The calculated degree of circular polarization (DCP) based on the PL decay curves shown in (b). The instrument response function (IRF) measured at ~ 2.6 eV is shown as a comparison (shaded grey). (d) The DCP obtained at three emission energies indicated by the black arrows in (a). The DCP is relatively low ($\sim 20\%$) because the PL decay curve is measured under a non-resonant excitation, which is consistent with the observations with the transient absorption spectra shown in Fig. 4b in the main text.

Supplementary Fig. S10. Room-temperature PL (red curve) and absorbance (blue curve) spectra of CsPbCl₃ (a) and CsPbBr₃ (b) thin films. The PL decay curves with the $\sigma^+\sigma^+$ (blue curve) and $\sigma^+\sigma^-$ (red curve) excitation-detection polarization configuration at ~ 2.99 eV for the CsPbCl₃ (c) and at ~ 2.36 eV for CsPbBr₃ (d) thin films.

In contrary, a very long rising time (~ 400 ps) is observed in the PL kinetics of CsPbCl₃ thin films (Fig. S10c). This indicates that the conversion process from the absorption to emission states are very slow. Therefore, excitons lost their spin polarization within this long conversion process. For CsPbBr₃ thin films, we didn't see the exciton spin dynamic in the PL decay curves, which is because the excitation is too far away from the exciton states and initially leads to a very low spin injection efficiency of excitons. We think the TRPL results are very exciting and worth further investigations to understand the transient exciton spin properties and distinct emission and lattice dynamics in halide perovskites. In the future, we will seek collaborations to study the emission dynamics in CsPbX₃ thin films by using setups with the sub-ps temporal resolution (such as by using Streak camera and fluorescence upconversion spectrometer) and using excitations with tunable wavelength.

Minor issues:

[5] The basic characterizations of the single crystal form of the prepared thin film are missing in the S.I., such as XRD, roughness and surface morphology by the AFM (not the only the 'profilometer'), particularly the circularly dichroism (CD) spectra for the thin film without the pump.

Response: We thank the reviewer's kind suggestions and provided the fluorescence image, AFM, SEM and XRD characterizations in Fig. S1 of the Supplementary Information, and the TEM characterization of CsPbBr₃ thin films is also discussed in the response of Comments (1) raised by Reviewer1. There were no circularly dichroism spectra for the samples without the pump (Fig. S5).

[6] In Fig. 3c, the olive square represents the decay at 89.1 uJ/cm^2 . But there are only olive diamonds shown in the figure. The y-axis would be better in the unit of circularly polarization degree.

Response: We thank the reviewer for spotting this issue and have revised it accordingly. We did not change the unit of y-axis for simplicity since it does not affect the extraction of spin decay times.

[7] In Fig. 3b, the lifted spin degeneracy for the spin up and spin down exciton states induced by the circularly polarized pump is not presented.

Response: We thank the reviewer for this kind suggestion and revised the schematics in the current version of the manuscript.

Reviewer #3:

In this work the spin dynamics of photoexcitations in the popular inorganic perovskite semiconductors, namely CsPbBr_3 has been studied using the transient circularly polarized pump probe technique at various excitation intensities and temperatures. The experimental part is not new in the field. Spin dynamics using the same technique has been studied in various 2D and 3D hybrid organic inorganic perovskites, including by the authors. Thus there is no novelty here.

Response: We greatly appreciated the Reviewer's constructive comments and suggestions which helps us to improve the quality of the manuscript substantially. We also apologize that the novelty and significance of our observations may not have been adequately highlighted in the previous version of our manuscript, which could have affected the reviewer's view about our work. Firstly, despite of a number of publications reporting the spin-dependent properties of hybrid organic-inorganic perovskites in the past few years, this field is still at its very early stage. Critical questions, such as the origin of the spin coherence and spin relaxation mechanism, are greatly in dispute as discussed in Paragraph 2 at Page 6 in the main text. Secondly, the significance and novelty of our work lie in the observation of **long exciton spin lifetime** and **surprisingly strong many-body interactions of spin-polarized excitons in bulk all-inorganic perovskites**. For the first time, we reported:

- 1) The exciton spin relaxation time in CsPbX_3 is up to ~ 20 picoseconds (ps) at low pump fluence, which is the longest reported in the metal halides perovskites family at room temperature.
- 2) The pseudo-magnetic field arising from spin-dependent many-body interaction of excitons is huge, and can lead to a giant spin splitting of ~ 12 meV at a moderate pump

fluence of $\sim 9.4 \mu\text{J}/\text{cm}^2$. Such distinct photoinduced optical nonlinearities in bulk CsPbX_3 are surprisingly strong compared with those of two-dimensional (2D) semiconductors, such as the 2D perovskites, GaAs-based quantum wells and 2D transition metal dichalcogenides. We strongly believe that these significant observations have a profound bearing on fundamental understanding of the spin physics of halide perovskite semiconductor and have strong implications for room-temperature spintronics and quantum optics devices based on halide perovskites. We hope our justifications presented above could satisfy the Reviewer's concerns.

In addition the message of this work is not proven unambiguously. Firstly there is no experimental proof that the circular polarized bands that are separated by the giant energy of few tens of meV belong to the same state to begin with. In addition, authors mention many-body interaction as a mechanism for the giant photoinduced circular dichroism, but have not done any calculation to show it. This is absolutely needed for publication in high profile journal such as Nature Commun. Authors mention other phenomena such as bandgap renormalization and state filling, but have not tried to use these mechanisms to fit the obtained polarized spectra. Thus the interpretation of the results falls in the realm of speculation.

Response: We thank the Reviewer for his/her critical comments and suggestions. We are sorry for the confusions caused due to the lack of adequate discussions on the possible electronic phase transition between the exciton and free-carrier states (i.e., the excitonic Mott transition). First of all, we can simply calculate the Mott density (n_M) based on the Mott criteria [$(n_M)^{1/3} \cdot a_B \approx 0.2$, where a_B is the exciton Bohr radius] [Mott N. F., *Rev. Mod. Phys.* 40, 677 (1968)]. The n_M is $\sim 1.8 \times 10^{17} \text{ cm}^{-3}$ with an exciton Bohr radius of 3.5 nm in bulk CsPbBr_3 (Ref. 22). However, the Mott criteria is derived based on the screening effect from free carriers. Here we directly inject excitons in the system by using resonant excitations. The screening effect from these charge neutral quasiparticles (exciton) is about one order of magnitude weaker than that of free carriers (Ref. 45). Therefore, the actual Mott density could be $\sim 1.8 \times 10^{18}$ in CsPbBr_3 , which is corresponding to a pump fluence of $\sim 31.1 \mu\text{J}/\text{cm}^2$ if assuming that the absorption of the pump beam is in the linear regime. We can safely claim that the spin splitting shown in Fig. 2c in the main text are indeed primarily from the many-body interaction of spin-polarized excitons with pump fluence below $\sim 10 \mu\text{J}/\text{cm}^2$. The energy splitting between +1 and -1 exciton states, as well as the bleaching of the exciton absorption, show a linear increase with the increase of pump fluence ($< 10 \mu\text{J}/\text{cm}^2$), which further support our claims.

Secondly, the above analysis is undoubtedly supported by the recent experimental and theoretical results presented in Ref. 10 [Yang Y. et al., *Nat. Photon.* 10, 53–59(2016)]. Yang et al. studied the room-temperature TA spectra of methylammonium lead iodide (MAPbI_3) polycrystalline films. Despite of a small exciton binding energy of $\sim 9 \text{ meV}$ in MAPbI_3 , Yang et al.

revealed unambiguously that the saturation effect and the excitonic Mott transition happen at a large excitation density of $\sim 1 \times 10^{18} \text{ cm}^{-3}$ at which the bandgap renormalization of the continuum band overwhelms the excitons binding energy. Therefore, the excitonic Mott transition would occur at an excitation density larger than $\sim 1 \times 10^{18} \text{ cm}^{-3}$ in CsPbBr₃ owing to the large exciton binding energy of $\sim 40 \text{ meV}$.

We thank the Reviewer for his/her kind suggestions to conducted theoretical calculations to quantitatively understand the optical nonlinearities and many-body interactions of spin-polarized excitons in CsPbX₃. We believe that the major findings about the 20 ps spin relaxation time at low excitation densities and large spin splitting of excitons are directly manifested by the experimental data. By performing theoretical calculation similar to those in Ref. 10 and Ref. 38, the strength of excitonic many-body interaction can be quantified, and more specifically the contribution from excitons and/or free carriers at high pump fluence would be distinguished. Nevertheless, these perspective theoretical results would not affect the major conclusions mentioned above in this manuscript. Therefore, we would like to conduct theoretical calculations to get more fundamental insights in the future as a different project.

We have added the above discussions in the revised Supplementary Information (Page 4-5).

Also authors refer several times papers on GaAs MQW from the 90th, where the relation between the exciton and band-edge carriers was not recognized. In the meantime the inter-relation between excitons and the continuum in the absorption spectrum of a semiconductor, especially for relatively small exciton binding energy such as in CsPbBr₃ has been advanced. Authors have not taken this into account in their discussion.

Response: We thank the reviewer for the comment. We agree with the Reviewer that the exciton binding energies in GaAs quantum wells are small and usually below the thermal energy at room temperature (i.e., $\sim 26 \text{ meV}$). However, the optical properties and many-body interactions of excitons in GaAs quantum wells are well understood both experimentally and theoretically in the past three decades. We cite the papers on GaAs quantum wells from the 1990s because these were pioneering work and widely recognized in the fields of semiconductor optics, many-body physics and spintronics. For example, Ref. 33 [Stark J.B. et al., *Phys. Rev. B* 46, 7919 (1992)] demonstrated the nicely resolved exciton Rydberg series and the spin-dependent many-body interaction of excitons in GaAs quantum wells at liquid helium temperature (4K). And Ref. 48 [Maialle M. Z. et al., *Phys. Rev. B* 47, 15776 (1993)] presented very detailed theoretical calculation on the exciton spin dynamics and spin relaxation mechanism based on the GaAs quantum wells. Nevertheless, these theories can be universally applied to other semiconductor materials with similar electronic structure and spin degeneracy at the band edge, like the CsPbX₃ samples.

Following the Reviewer's suggestion, we have provided detailed discussion about the relation between the exciton and the continuum band and possible electronic phase transition under optical excitation as discussed above and also in Page 4-5 of the revised Supplementary Information.

For these reasons I do not recommend publication of this work in Nature Commun.

Response: Following the comments and suggestions from all the Reviewers, we have provided supplementary experiments and more data analysis, and revised the manuscript accordingly. We believe that the quality of our manuscript has been improved substantially and thus merits publication in Nature Communications.

REVIEWERS' COMMENTS

Reviewer #1 (Remarks to the Author):

The authors have addressed this reviewers comments and I can recommend publication of the results.

Reviewer #2 (Remarks to the Author):

The authors have made significant efforts to adequately answer all the raised issues in the revised manuscript and further tighten their claims by making additional experiments and analysis. In the revised work, the transient polarization dependent photoluminescence (PL) measurement has been succeeded in the CsPbX₃ single crystals which has been proven very challenging at room temperature. Although the dynamics of spin-polarized excitons are still limited by the temporal resolution, the fingerprints of spin characters in the transient PL emission are clear and promising. This supports the proposed model. Therefore, I recommend publication of this revised manuscript in Nature Communications.

Some questions for consideration:

The control experiments of the transient absorption using left-handed polarized excitation are consistent with the authors' statement. I would suggest moving to Fig. S2 as one of the panels in Fig. 1, which may manifest a reversed case of circularly polarization excitation for the photoinduced circular dichroism. This may eliminate the doubts about the structure-induced static or dynamic circular dichroism in the CsPbBr₃ single crystals. Has the pump fluence dependence of circular polarization degrees been measured using the left-handed circularly polarized excitation? What's the obtained ΔE at 0.3 ps at a low pump fluence?

I am curious about the role of predicted pronounced Rashba effect in this material [ref. 22]. Is this long spin lifetime attributed to the indirect bandgap induced by the Rashba effect?

Point-to-Point Response to Reviewer's Comments:

Reviewer #1:

The authors have addressed this reviewer's comments and I can recommend publication of the results.

Response: We greatly appreciate the Reviewer's recommendation for publication of our work.

Reviewer #2:

The authors have made significant efforts to adequately answer all the raised issues in the revised manuscript and further tighten their claims by making additional experiments and analysis. In the revised work, the transient polarization dependent photoluminescence (PL) measurement has been succeeded in the CsPbX₃ single crystals which has been proven very challenging at room temperature. Although the dynamics of spin-polarized excitons are still limited by the temporal resolution, the fingerprints of spin characters in the transient PL emission are clear and promising. This supports the proposed model. Therefore, I recommend publication of this revised manuscript in Nature Communications.

Response: We greatly appreciate the Reviewer's recommendation for publication of our work. We have carefully considered the constructive suggestions from the Reviewer and provided supplementary results in the revised manuscript. We believe that the remaining concerns are all addressed unambiguously.

Some questions for consideration:

The control experiments of the transient absorption using left-handed polarized excitation are consistent with the authors' statement. I would suggest moving to Fig. S2 as one of the panels in Fig. 1, which may manifest a reversed case of circularly polarization excitation for the photoinduced circular dichroism. This may eliminate the doubts about the structure-induced static or dynamic circular dichroism in the CsPbBr₃ single crystals. Has the pump fluence dependence of circular polarization degrees been measured using the left-handed circularly polarized excitation? What's the obtained ΔE at 0.3 ps at a low pump fluence?

Response: We thank the Reviewer's constructive suggestions. We have conducted the pump fluence dependent TA measurements with the left-handed circularly polarized excitation. The results are presented in the Figure shown below (i.e., the updated Supplementary Figure 2). There is no notable difference between the σ - σ - (σ - σ +) and σ + σ + (σ + σ -) TA spectra of CsPbBr₃ as a function of pump fluence. Therefore, we would like to keep Supplementary Figure 2 in the Supplementary Information in order to have a concise Fig. 1. These results indeed help to eliminate possible contributions from the structure-induced static or dynamic circular dichroism in single crystalline CsPbBr₃. The obtained ΔE at 0.3 ps at a low pump fluence of 0.9

$\mu\text{J}/\text{cm}^2$, for instance, is ~ 1.9 meV using the left-handed circularly polarized excitation, the same as that using the right-handed circularly polarized excitation.

Supplementary Figure 2. The TA spectra of CsPbBr₃ obtained with different pump-probe configurations at 0.3 ps at room temperature. **a-c** The pump fluence are ~ 0.9 , 4.5 and 9.0 $\mu\text{J}/\text{cm}^2$, respectively. The photon energy of pump beam is ~ 2.4 eV, which is in resonant with the exciton states. The differences between the $\sigma\text{-}\sigma\text{-}$ ($\sigma\text{-}\sigma\text{+}$) and $\sigma\text{+}\sigma\text{+}$ ($\sigma\text{+}\sigma\text{-}$) TA spectra are less than 5% at high pump fluence and varies randomly in multiple measurements, which is caused by the error ($\pm 0.5^\circ$) of manually rotated quarter-waveplates.

I am curious about the role of predicted pronounced Rashba effect in this material [ref. 22]. Is this long spin lifetime attributed to the indirect bandgap induced by the Rashba effect?

Response: In ref. 22 [“Rashba Effect in a Single Colloidal CsPbBr₃ Perovskite Nanocrystal Detected by Magneto-Optical Measurements”. Nano Letters 17, 5020, (2017)], the authors discussed the Rashba effect in CsPbBr₃ nanocrystals because of the spin-orbital coupling and a lack of inversion symmetry. Such inversion symmetry breaking is mainly from the surface effect in CsPbBr₃ nanocrystals. On the contrary, the inversion symmetry preserves in bulk CsPbBr₃, and there is no Rashba effect [ref. 15 and 22]. The long exciton spin lifetime in bulk CsPbBr₃ results from an adequate exciton binding energy (~ 40 meV), which generates an excellent balance of the robust exciton resonance at room temperature and relatively slow spin depolarization controlled by the BAP mechanism.